# Online and Offline Aggressive Behaviors in Adolescence: The Role of Self-Regulatory Self-Efficacy Beliefs

**DOI:** 10.3390/bs14090776

**Published:** 2024-09-04

**Authors:** Ainzara Favini, Carolina Lunetti, Alessia Teresa Virzì, Loreta Cannito, Flavia Culcasi, Tiziana Quarto, Paola Palladino

**Affiliations:** 1Department of Humanities, University of Foggia, 71121 Foggia, Italy; tiziana.quarto@unifg.it (T.Q.); paola.palladino@unifg.it (P.P.); 2Faculty of Education Sciences, Guglielmo Marconi University, 00193 Rome, Italy; c.lunetti@unimarconi.it; 3Department of Psychology, Sapienza University of Rome, 00185 Rome, Italy; alessia.virzi@uniroma1.it; 4Department of Social Sciences, University of Foggia, 71121 Foggia, Italy; loreta.cannito@unifg.it; 5Clinic for Substance and Behavioral Addiction, Academic Foundation Policlinic Agostino Gemelli IRCCS, 00168 Rome, Italy; flaviaculcasi@gmail.com

**Keywords:** self-regulatory self-efficacy, online aggressions, aggressive behaviors, impulsivity, adolescence

## Abstract

Self-regulatory self-efficacy belief (i.e., SRSE) represents a fundamental factor for adjustment in adolescence, as a vehicle to promote positive behaviors and protect youths from transgressions and maladjustment. Research attested that, during adolescence, boys are more vulnerable to externalizing behaviors than girls, especially when they perceive themselves as scarcely capable of managing and orienting their behaviors and when they possess impairments in impulsivity. Previous studies firmly supported the crucial role of SRSE, especially in the offline context in adolescence. Still, very few studies investigated its impact in the online context, although nowadays, the Internet represents one of the most significant environments for youths’ daily lives. Thus, we aimed to examine the protective moderating role of SRSE in online and offline aggressive behaviors beyond youths’ temperamental vulnerabilities, such as high impulsivity. A sample of 318 Italian adolescents (M_age_ = 15.21; *SD* = 0.51; 57% boys; 40% girls; 3% third gender) were asked to complete the Impulsivity Scale at Wave 1, online and offline aggressive behaviors scales at Wave 2, and SRSE at Wave 2. The multiple-group-by-gender moderation model evidenced that, for what concerns online aggression, beyond the direct effects of impulsivity in boys and girls, SRSE directly affected online aggression and mediated the effect of impulsivity in girls. In contrast, impulsivity significantly affected offline aggressive behaviors only indirectly through the impact of SRSE, and SRSE directly influenced these behaviors in both genders. These results hold for the effects of youth’s age, sexual orientation, socioeconomic status, and years of education completed. This work preliminary evidenced that, contrary to previous studies which focused mainly on the vulnerability of boys to aggressive conduct, impulsivity had direct effects on online aggressive behaviors in girls, and SRSE can protect adolescents of both genders in the online context by predicting low online aggression and represents a protective factor from the indirect effects that impulsivity impairments can also have in the offline context.

## 1. Introduction

According to a socio-cognitive perspective of individual development and functioning [1,2], individual characteristics, environmental characteristics, and behaviors are strictly interconnected in predicting positive or negative developmental pathways [3]. This perspective emphasized the moderating role that individuals’, and youths’, beliefs and reasonings can have in the relation between individual characteristics, such as dispositional susceptibilities, temperamental impairments, or personality vulnerabilities [4,5,6], and the development of maladaptive behavioral responses, such as oppositive, antisocial or aggressive conducts [5,7]. Despite this topic being extensively studied in offline contexts within adolescent populations [5,8,9], very few studies examined the role of personal beliefs in the associations between individual susceptibility, irritability, and aggressive behaviors in online contexts [2,10], as this topic is still relatively new. Most of the existing studies focused on self-efficacy beliefs specifically related to individuals’ perceived ability to use the Internet and technological devices rather than exploring their perceived competence of controlling and effortfully directing their behaviors while navigating social media [11,12,13,14,15]. Therefore, the present study aimed to overcome this gap by investigating the concurrent and differential role of self-regulatory self-efficacy beliefs as mediators of the associations between impulsivity and online/offline aggressive behaviors in adolescent boys and girls, considering their gender and background characteristics. 

### 1.1. Online and Offline Aggressions in Adolescence: Individual and Behavioral Factors

Impulsivity is a core temperamental and formal characteristic of human functioning, reflecting the rapidity/slowness of responding to internal or external stimulation in neutral or provoking conditions. It is also associated with unplanned actions and a lack of evaluations of the consequences of personal actions [8,9,16]. This tendency is associated with various related behavioral processes, such as risk-taking, inadequate decision-making and problem-solving, high sensation-seeking, and hedonic well-being [7,9,17,18]. Adolescence’s developmental period is interesting for this tendency because adolescents are normatively more inclined to engage in risky behavior, experience heightened emotional susceptibility, and face challenges in activating self-regulating skills [7,19,20,21,22].

Associations between impulsivity and aggressive behaviors are widely documented [8,9,23,24]. Cognitive perspectives and social-information processing models [25,26] underlined how hostile cognitive schemas in impulsive children and adolescents can exacerbate their probability of engaging in aggressive responses [16,27,28,29]. Thus, youths are more inclined to behave impulsively and struggle to evaluate the possible negative consequences of their actions. They also show less cognitive and self-regulatory abilities to inhibit predominant behavioral responses, preferring a more adaptive behavior [16,30]. 

In this scenario, research has established how gender plays a crucial role in individuals’ adaptive or maladaptive behavioral responses [31,32,33]. Young girls tend to express more covert aggressive behaviors, while young boys are more likely to show overt and manifested aggression [8,27]. These gender differences can be attributed to the different cognitive processes associated with each type of aggressive response. On average, girls are more inclined to evaluate the consequences of their actions before acting and experience guilt and shame as negative emotions in response to aggressive behavior. In contrast, boys are generally more inclined to an immediate behavioral response (due to a higher loss of control over their actions and behaviors), which leads them to behave more impulsively and aggressively than girls [29,34]. 

Mechanisms that connect impulsive tendencies to aggressive behaviors are quite similar in offline and online contexts [35,36,37]. Impulsive tendencies, especially concerning low control of internal stimuli and impulses, have been firmly linked to high involvement in aggressive and deviant behaviors in offline and online environments [10,17,30]. Moreover, classical perspectives suggest that impulsivity (which pertains to low self-control, high stimuli sensitivity, the tendency to interpret external stimulation as potentially harmful, and impairments in adaptive emotion and behavioral regulation) may predispose individuals—especially younger ones—to react aggressively, independently from the contexts [38,39,40]. Impulsive people are likely to engage in aggressive behaviors in different offline contexts, such as in academic, work, or relational situations [30,41], and this behavioral pattern would replicate similarly in online contexts, such as while navigating on Social Networks [17,35,36]. However, newer perspectives, such as the online disinhibition effect theory or communication theories on computer-mediated-communications [42,43], have pointed out that online environments have peculiar characteristics that make them different from any other offline environment, such as the potential anonymity of perpetrators or the a-synchronicity of relational interactions between the perpetrator and the victim/s which did not allow the perpetrator to obtain an immediate response and reward from their actions. Therefore, it would be controversial to affirm that impulsivity would play a similar role in predicting offline and online aggressive conduct [10,39]. In addition, the limited existing studies that focused on the role of impulsivity in online aggression primarily focused on cyberbullying [36,44]. Even a smaller number of studies focused on other forms of online aggression, such as online hate, cyber-stalking, and engaging in “shitstorms” (i.e., a collective form of online aggression in which a group of people intentionally posts hateful and aggressive comments directed at a specific individual or social account) [45], as well as online sexual, violent harassment [10], swearing, trolling and flaming [46]. 

### 1.2. The Moderating Role of Self-Regulatory Self-Efficacy Beliefs

Adaptive self-regulatory abilities play a crucial role in helping youths modulate and manipulate their experiences. Individuals’ perceptions of their abilities to manipulate and control situations represent one of the most influential determinants of adjustment [1,3,47]. From a socio-cognitive perspective, Bandura conceptualized these perceptions as self-efficacy beliefs that represent “dynamic constructs that can be enhanced through mastery experiences as a result of individuals’ capacities to reflect and learn from experience” ([47] p. 1). Self-efficacy beliefs guide habits and tendencies, highlighting individuals’ proactive role in controlling their lives based on cognitive self-regulation and reflective thinking, which can also influence motivations and goal orientation [3]. These beliefs help individuals modulate their goal-oriented behaviors when faced with challenging situations [1,47]. Consequently, those who feel capable of activating their self-regulatory abilities are more likely to achieve their aspirations and objectives because these beliefs represent one of the most predictive success factors [1,3]. 

Self-efficacy beliefs are not a universal construct but vary according to the specific domain of functioning involved. Thus, individuals may possess different self-efficacy beliefs for various contexts perceived as challenging [48]. In this sense, the extent to which individuals feel adequately capable of regulating their behaviors toward transgressive activities or activating self-regulatory skills against peer pressure is an example of self-regulatory efficacy [1]. Previous studies attested that self-regulatory self-efficacy beliefs can serve as a bridge between individual and contextual influences on adaptive or maladaptive responses [49]. 

For youths, these capabilities are fundamental skills to adequately organize their behaviors toward others in offline and online contexts, such as in school settings or while navigating social networks [2,47,50,51,52]. Previous studies highlighted the importance of adequate levels of self-regulatory self-efficacy beliefs in protecting youths and adults from engaging in aggressive behaviors and antisocial conduct offline (see [53] an extensive meta-analysis). Adolescents with higher self-regulatory self-efficacy beliefs engage in more prosocial behaviors, experience greater psycho-social well-being, are less susceptible to internalizing and/or externalizing problems, perform better in school settings, and engage in less aggressive conduct [47,53,54]. Conversely, adolescents with low self-regulatory self-efficacy beliefs are more inclined to engage in risky activities, are more vulnerable to substance use and/or abuse, show academic problems, and tend to engage in more aggressive behaviors offline and online, including a greater tendency to engage in cyberbullying [54,55]. Thus, this duality highlights the importance of maintaining adequate levels of self-regulatory self-efficacy in youths, as it mitigates the effects of impulsivity on aggressive behaviors in both online and offline relational contexts and enhances their overall adjustment and resilience [53,55].

### 1.3. The Present Study

Based on the above-mentioned theoretical premises [2,10,53], research has well-attested the protective role of self-regulatory self-efficacy beliefs. However, very few studies have investigated these effects in online contexts [10,36]. Understanding these mechanisms in online contexts represents a crucial and challenging aim, given the widespread prevalence of online aggressive conduct, such as cyberbullying, hate speech, flaming, and so on [10,45,46]. Therefore, it is critical to understand these dynamics, especially for youths who are the most vulnerable target for engaging in aggressive conduct because of their higher sensitivity to negative emotions and lower self-regulatory abilities [56,57,58] but also because younger people use technological devices more extensively, making them more susceptible to incur in online risks [17,59]. When analyzing these associations in online and offline contexts, it is important also to consider the youths’ gender, as most research has evidenced crucial differences in engaging in aggressive behaviors in this regard [31,56]. 

Therefore, the general aim of the present study was to fill the gap in the literature by investigating the role of self-regulatory self-efficacy beliefs in mediating the effects of impulsivity in aggressive responses, not only in traditional offline contexts [8,9] but also analyzing the specific associations between impulsivity and aggressions in online settings [17]. To answer our general research question, we tested the potential mediating and protective role of self-regulatory self-efficacy, differently in the associations between impulsivity and offline aggression, and impulsivity and online aggression, and the kind of these associations in adolescent boys and girls, to analyze the moderating role of youths’ gender [56]. The following concrete and specific hypotheses were tested:

**H_1_.** 
*Regarding offline aggressive behaviors, we hypothesized that high impulsivity would predict higher aggressive responses in both adolescent boys and girls (H_1_ a), and, according to socio-cognitive theory, the protective role of self-regulatory self-efficacy beliefs would be stronger in boys who are the most vulnerable to these associations (H_1_ b) [3,53].*


**H_2_.** 
*Regarding online aggressive behaviors, our work was exploratory due to the limited literature on this topic. Still, according to the limited previous research, we would hypothesize an effect of impulsivity on online aggressions similar to that identified for offline aggressions [10,39]. More specifically, according to the online disinhibition theory and the computer-mediated-communication theory [42,43] that emphasized the specificity of the online context, which allows anonymity and broader space to externalize aggressive tendencies, we hypothesized a more substantial effect of impulsivity on online aggressive behaviors, compared with the offline counterpart of these associations (H_2_ a). We are not aware of previous studies that investigated the potential protective role of self-regulatory self-efficacy in the association between impulsivity and online aggression, but taking in mind that online aggression, compared with offline aggression, pertained to more girls, the protective role of self-efficacy beliefs could also emerge for adolescent girls, but this is an exploratory hypothesis (H_2_ b).*


## 2. Materials and Methods

### 2.1. Participants and Procedures

Participants were drawn from a wider longitudinal national project that was carried out in a junior high school located in Rome, which was a school-based intervention with the twofold aim of preventing online problematic behaviors while promoting positive behaviors in the online and offline social contexts [5]. For the purposes of the present study, we considered youths who completed both pre- and post-intervention assessments (i.e., Wave 1 and Wave 2). Before the data collection, informed consent was obtained from the parents of the youths involved in the project. Questionnaires were administered to each student during school hours, using an online platform to ensure the anonymity of each participant. At the beginning of each questionnaire, specific informed consents were collected. A total sample of 318 adolescents aged 14 to 18 years was considered (M_age_ = 15.21; *SD* = 0.51). Most youths in the present study enrolled in the second grade of junior high school (88% of the total sample). They were mostly on time with their academic pathways (only 1% of students repeated one or more years of instruction). With regards to their gender, youths were mostly distributed across the feminine (N = 156; 40% of the total sample) and the masculine (N = 225; 57% of the total sample) genders, and a small percentage of youths to the third gender (N = 11; 3% of the total sample). Youths mostly declared a heterosexual orientation (87% of the total sample), small percentages of other sexual orientations were registered (respectively, 1% homosexual, 4% bi-sexual, 3% fluid, and 5% of other LGB+ sexual orientations), and they mostly declared being single (70% of the total sample). 

Regarding socio-economic status, most of the youths involved in the study lived with spoused parents (79% of the sample), of whom 88% of mothers and 97% of fathers had a full- or part-time occupation. Parents mostly declared an average-to-high educational level (36% of mothers and 37% of fathers had a high school diploma, and 44% of mothers and 39% of fathers had a bachelor’s or a master’s degree). 

### 2.2. Measures

#### 2.2.1. Background Characteristics

Descriptive information about the sample, such as youths’ age, gender, sexual orientation, years of formal instruction completed, and socio-economic status, was collected in the first Wave. Gender was coded as 0 for adolescent boys and 1 for adolescent girls, and sexual orientation was coded as 1 for heterosexual, 2 for homosexual, 3 for bisexual, and 4 for other LGB+ orientations. Years of formal instruction completed were directly asked by each student and recorded considering five years of primary school, three years of the first grade of secondary school, and each year of the second grade of secondary school completed (i.e., for those who enrolled in the second year of junior high school, were considered 5 + 3 + 2). The socioeconomic status of youths involved in the study was computed considering parents’ education and work occupation. Detailed information about the correlations among the study variables is reported in Table A1. Information about continuous variables is provided below. 

#### 2.2.2. Self-Regulatory Self-Efficacy

To assess adolescents’ perception of their own self-efficacy beliefs regarding their capabilities to self-regulate and orient behaviors, we used six items derived from the Self-Regulatory Self-Efficacy Beliefs Scale [60,61], which assesses perceived abilities to resist peer pressure in engaging risky and transgressive behaviors, as well as to orient their behavior in a self-consciousness way to achieve planned goals [49,62]. Each item was rated on a 5-point Likert scale from 1, Not well at all, to 5, Very well (e.g., “How well can you resist peer pressure to do things that can get you in trouble?” or “How well can you avoid behaving in a transgressive manner even when the risk for a punishment is very limited?”). A larger body of research firmly establishes the psychometric properties of this scale, both cross-culturally and longitudinally, across different phases of adolescence [61]. In our study, internal consistency was good (ω = 0.843 and α = 0.842; see Table A1).

#### 2.2.3. Impulsivity

Youths’ self-evaluations of their impulsivity levels were assessed at Wave 1, adopting the Barratt Impulsiveness Scale–Brief [18,63]. The scale was developed and used to measure different aspects of impulsivity, such as difficulties in self-regulatory processes, motor and attentive impulsivity, and lack of perseverance. Each item was rated on a 4-point Likert scale, from 1, Never, to 4, Almost always/always (e.g., “I do things without thinking” or “I don’t pay attention”). This scale is one of the most used worldwide to assess impulsivity levels, and a variety of previous research supported the validity of this instrument [63,64]. In our study, internal consistency was acceptable (ω = 0.710 and α = 0.772; see Table A1).

#### 2.2.4. Online and Offline Aggressive Behaviors

Two different scales were considered to assess online and offline aggressive behaviors at Wave 2. To measure offline aggression, we used five items derived from the Youth Self-Report (YSR; [65]), which assesses the type of aggressive behaviors acted by the individual within the last six months, using a 3-point scale ranging from 0 “Not true,” to 2 “Very often true” (e.g., “I am cruelty, I bullied, or meanness to others,” “I physically attack people”). To measure online aggressive behaviors, we used the Online Aggression Scale [66], a four-item instrument developed to assess a variety of online aggressive behaviors acted by the individual within the last 30 days, such as threatening others, insulting or stalking other people, each of them rated on a 4-point Likert scale ranging from 0 “Never,” to 3 “Very often” (e.g., “Make rude or nasty comments about someone else online,” or “Use the Internet to threaten or embarrass someone”). Both measures did not specifically distinguish between overt and covert aggression, and items derived from the YSR for the offline aggression captured primarily relational forms of aggressive behaviors, both verbal and physical, while the measure used for the online aggressions captured exceptionally verbal and indirect aggressive behaviors. The YSR measure was broadly and widely adopted worldwide, and there is strong evidence of its cross-cultural and longitudinal validity [67,68]. In our study, internal consistency was good (ω = 0.916 and α = 0.911; see Table A1). The Online Aggression Scale was a relatively new instrument and was adopted especially in Asian countries [37,69], so no Italian validation of the instrument is available to our knowledge. Despite these limitations, in our study, the internal consistency of this measure was good (ω = 0.814 and α = 0.804; see Table A1).

### 2.3. Statistical Approach

All analyses were run within Mplus 8.11, and to test our hypotheses, we adopted the following steps. We preliminary confirmed any significant influence of the school-based intervention that our participants followed as a part of the broader project from which we selected our sample.

First, we examined the relations among the study variables in the general sample using a simple mediation model, considering impulsivity as the direct predictor of online and offline aggressive behaviors and self-regulatory self-efficacy beliefs as the possible mediator of these associations [70]. 

Then, we analyzed these relations in separate gender groups. Considering the smaller percentage of the third gender in our sample (3% of the total sample), we only considered the masculine and feminine genders. So, we tested our models separately in adolescent boys and girls, running a multiple-group mediation model, considering youths’ gender as a grouping variable, and controlling for youths’ sexual orientation, age, socioeconomic status, and the years of formal instruction completed by students [71]. 

We used Robust Maximum Likelihood Estimation (MLR) for continuous variables [72] and considered the following criteria to evaluate the goodness of fit: χ^2^ Likelihood Ratio Statistic, the Comparative-Fit Index (CFI), and the Tucker–Lewis-Fit Index (TLI) greater than 0.95 [73], the Root Mean Square Error of Approximation (RMSEA) with associated confidence intervals lower than 0.05, and the Standardized Root Mean Square Residual (SRMR) lower than 0.06 [74]. We first ran a model in which we fully constrained all the parameters to be equal across groups and then a model in which we freely estimated all the parameters, comparing these two models using the chi-square difference test [74]. We released one constraint per comparison until the chi-square difference test showed a non-significant increase in the chi-square, adopting a cutoff for the significance of *p* < 0.01 (given that obtaining a significant chi-square becomes increasingly likely with large sample sizes [74]).

## 3. Results

Preliminary descriptive and exploratory analyses were adopted on all the study variables to investigate means and standard deviations, skewness and kurtosis, internal consistency of the utilized constructs, and correlations among all the variables. Detailed information on these procedures is provided in Table A1 and Table A2. 

### 3.1. Mediation–Moderation Model in the Full Sample

As the first step of our statistical approach, we ran the proposed mediation–moderation SEM model in the full sample to analyze the hypothesized associations in the whole sample, controlling for youths’ age, sexual orientation, socioeconomic status, and years of formal instruction completed by students [70,71]. The results of this model are shown in Figure 1.

This model showed an adequate fit [χ^2^ (*Df* = 295) = 358.209, *p* < 0.005; RMSEA = 0.026 (C.I. = 0.014–0.035); CFI = 0.974, TLI = 0.969; SRMR = 0.050], as shown in Table 1. 

For what concerns direct effects, higher impulsivity directly predicted higher aggression only in the online context (*β* = 0.16; *p* < 0.01), and self-regulatory self-efficacy beliefs predicted both lower online (*β* = −0.38; *p* < 0.001) and offline (*β* = −0.35; *p* < 0.001) aggression. With regards to indirect effects, impulsivity indirectly predicted both online (*β* = 0.15; *p* < 0.001) and offline aggression (*β* = 0.14; *p* < 0.001) through the effects of self-regulatory self-efficacy. 

### 3.2. Mediation–Moderation Models in Adolescent Boys and Girls

As the second step of our analyses, we estimated the same model examined in the previous paragraph within a multiple-group framework to analyze the emerged associations separately in the adolescent boys’ and adolescent girls’ samples to test whether there were possible differences in direct and/or indirect effects emerged in the full sample model [71].

The models estimated separately in the boys [χ^2^ (*Df* = 295) = 400.366, *p* < 0.001; RMSEA = 0.044 (C.I. = 0.033–0.055); CFI = 0.932, TLI = 0.920; SRMR = 0.065], and in the girl’s sample [χ^2^ (*Df* = 295) = 347.693, *p* = n.s.; RMSEA = 0.037 (C.I. = 0.016–0.052); CFI = 0.951, TLI = 0.943; SRMR = 0.080], showed an adequate fit, especially in the girl’s sample (see Table 1). Thus, we then estimated the multiple-group model in which we freely estimated all the parameters [χ^2^ (*Df* = 628) = 872.141, *p* < 0.001; RMSEA = 0.050 (C.I. = 0.042–0.058); CFI = 0.907, TLI = 0.898; SRMR = 0.080], to compare it with a nested model, in which we constrained all the parameters to be equal across the two groups [χ^2^ (*Df* = 657) = 912.119, *p* < 0.001; RMSEA = 0.049 (C.I. = 0.041–0.057); CFI = 0.906, TLI = 0.901; SRMR = 0.096]. The chi-square difference test [74] revealed that several parameters should be released across the two groups [χ^2^ diff (25) = 39.247; *p* = 0.034], as reported in Table 1. Therefore, considering the modification indices values, we released one parameter per time to compare the model with the previous one until this difference test became non-significant for *p* values higher than 0.05 [74]. The final multi-group mediation–moderation model [χ^2^ (Df = 654) = 902.634, *p* < 0.001; RMSEA = 0.050 (C.I. = 0.041–0.057); CFI = 0.906, TLI = 0.901; SRMR = 0.096] reported an adequate fit. In particular, we freely estimated the indirect effect of impulsivity on online aggression, the indirect effect of impulsivity on offline aggression, and a correlation between the second and fifth items of the self-regulatory self-efficacy scale. The results of this procedure are reported in Table 1, and the final model is reported in Figure 2.

As shown in Figure 2, we did not find significant direct effects from impulsivity to offline aggression, confirming this result in the full sample model. Direct effects from impulsivity to online aggression were significant only for adolescent girls (*β* = −0.27; *p* < 0.05). Direct effects of self-regulatory self-efficacy to offline (*β_boys_* = −0.35; *p* < 0.001; *β_girls_* = −0.38; *p* < 0.001) and online (*β_boys_* = −0.25; *p* < 0.001; *β_girls_* = −0.51; *p* < 0.001) aggression were confirmed and were equal across the two groups. 

With regards to indirect effects (see Table 2), impulsivity indirectly predicted both online (*β* = 0.22; *p* < 0.001) and offline aggression (*β* = 0.16; *p* < 0.001) through the effects of self-regulatory self-efficacy only for adolescent girls, while the protective role of self-regulatory self-efficacy beliefs was significant in adolescent boys only in the offline context (*β* = 0.13; *p* < 0.01).

## 4. Discussion

The present study contributed to filling the existing gap in the literature that extensively focused on associations among individual characteristics (i.e., impulsivity and self-efficacy beliefs) in predicting transgressive and maladaptive behaviors in offline contexts [9,53,54] and understudied these effects on online contexts [17]. In particular, we examined whether the associations between impulsivity and aggressive behaviors would be mitigated by the impact of self-regulatory self-efficacy beliefs, similarly or differently in online and offline contexts, and if, in these relations, the gender of the youths involved may play a role [8,34]. Our preliminary evidence supported self-efficacy’s differential role in reducing online and offline aggressive behaviors and its mediating role in the relationship between temperamental impulsivity and online aggressions in adolescent girls [2,10]. 

Regarding offline aggressive behaviors (H_1_ a), our results did not evidence any significant and direct association between impulsivity and offline aggressive conduct in the full sample, nor did we consider the moderating role of gender. This result was contrary to our hypothesis, which contemplated the effect of impulsivity on higher offline aggression in boys and girls. This result was also contrary to the large body of previous studies that underlined how higher impulsivity predicts higher aggressive behaviors in offline contexts, such as in school settings or at-home relational exchanges [41,75]. Thus, we found a significant indirect effect of impulsivity on offline aggressive behaviors through the impact of self-regulatory self-efficacy beliefs (H_1_ b), both in the full sample and when we considered the moderating role of gender, which was more robust in adolescent girls than boys. Thus, in our sample, adolescent boys and girls with higher impulsivity, especially regarding difficulties in emotion regulation and delay-discounting, two of the most salient characteristics of impulsivity, did not directly show higher aggressive tendencies in offline contexts, such as in relational or school settings [30,41]. However, when adolescents manifested serious difficulties in regulating their emotions and behaviors and showed impairments in coping with potentially aroused and ambiguous stimuli but at the same time felt sufficiently capable of effectively regulating their behaviors adaptively, they tended to act less aggressively rather than when they thought them as not adequately capable of regulating their behavior [47,54,55]. Unexpectedly, and contrary to our hypothesis (H_1_ b) that contemplated a stronger effect on boys who are more vulnerable to the impact of impulsivity on aggression, this pattern is especially true in adolescent girls rather than boys. We reasoned that this result could be ascribable to the low mean levels of offline aggressive behaviors in our sample, which in turn could not represent a concrete risk in our sample, so there would be any reasons to activate individual beliefs of behavioral agency to contrast temperamental impairments regarding impulsive tendencies [3,49]. Moreover, this absence of a significant direct association between impulsivity and offline aggressive tendencies could be ascribable to the strengthens of the role of self-regulatory self-efficacy, which pertained especially to the regulation of behaviors in offline social situations, and therefore could invalidate the negative effects of impulsive impairments on offline aggressive tendencies [1,3,47,49].

With regards to online aggressive behaviors (H_2_ a), overall, according to our hypothesis and to those theoretical approaches that investigated aggressive conduct online [42,43], we found stronger associations between impulsivity and these forms of aggression and a stronger mediating role of self-regulatory self-efficacy beliefs (H_2_ b), compared with associations found regarding offline aggression. In the full sample, we found a significant direct effect of impulsivity on online aggression, and, analyzing the moderating role of adolescents’ gender, that pattern was deeply identified as typical in adolescent girls rather than in their male counterparts (H_2_ a). This difference between the results in the full sample and the results of the multiple-group comparisons could be ascribable to the limited sample size, which could affect the strength of the association between impulsivity and online aggression [76]. Thus, in our sample, impairments in impulsivity predispose, especially adolescent girls, to be more inclined to act aggressively while they are online, such as using social to threaten or embarrass someone or making rude/nasty comments about others on social media [10,46,66]. Also, in this case, self-regulatory self-efficacy beliefs strongly directly reduce these online aggressive tendencies in both genders. In addition, adolescent girls who showed impulsive tendencies but who possess adequate self-regulatory self-efficacy beliefs were more protected from engaging in online aggressive conduct, according to our hypothesis (H_2_ b), and as proof of the crucial buffering role of individual self-efficacy beliefs in protecting people, especially the younger, from behavioral dysregulations [1,47,52]. Additionally, we confirmed that the effect of impulsivity on online aggressive behaviors was stronger, compared to offline aggression, due to specific characteristics of the context, such as anonymity and the online disinhibition effect [42,43]. It was captivating that these results came out regarding two different forms of aggression, online and offline, that we addressed adopting different time scales (i.e., for offline context, we asked about participants’ aggressive behaviors engaged in the last six months, whether for online context the timing was shorter, i.e., the previous month). We reasoned that the differential power of self-efficacy could be influenced in some way from the temporal aspect as a construct that reflects the dynamic nature of behaviors across time and that could be empowered through mastery experiences [3,48,53]. As a result, youths involved in our study could be influenced in their responses by considering especially situations in which they activated their agentic power that were closer to the moment in which the questionnaire was administered (i.e., in the last month, in the online context), rather than remote situations happened so far in the offline context [48,53]. 

Overall, these results supported the protective role of individual agency as a vehicle to support adaptive adolescents’ development rather than considering only emotional and/or behavioral difficulties, which may increase their risk of incurring maladaptive developmental patterns [16,22,25]. 

### Limitations

Our work represents an important step in understanding individual mechanisms that lead adolescents to behave aggressively, but we had to evidence several limitations of this study.

For one, we referred only to adolescents’ self-evaluation of their impulsive tendencies, self-efficacy beliefs, and aggressive behaviors. No other sources of information, such as the perception of teachers or their parents, were considered. Also, we could not include any genetic or neurophysiological indicators of impulsivity, despite mechanisms involved in dopaminergic circuitry that have been shown to undergo epigenetic modification in people who are exposed to social networks and the Internet and play an essential role in impulsive responses [18,77,78]. Several studies supported the view of aggressive behaviors as complex behavioral responses, of which younger people frequently did not have a clear and exhaustive view, as the sensitivity of those behaviors for social desirability and the uniqueness of individual points of view [32,79,80]. Therefore, future research should include other informants of youths’ behavioral responses to have a more fine-grained picture of aggressive tendencies and behaviors, online and offline. 

The second limitation of the study was related to the specific instruments included in our study. Regarding the self-regulatory self-efficacy measure, we only registered the kind of youths’ perception of their own beliefs in adaptively regulating behaviors and resisting peer pressure. We did not consider any related forms of regulatory abilities and beliefs, such as how frequently participants had to activate their self-regulatory skills or temperamental self-regulatory levels, such as effortful control or attentional focusing [4,47,75,81]. Regarding the aggression measures, we adopted measures of the two forms of aggression on different time scales (i.e., six months for offline aggression and one month for online aggression). We did not consider specific forms of overt or covert aggression or new online aggressive behaviors such as hate speech or trolling [45,46]. Future research could address these measurement limits by considering a unique timing for different forms of behavioral responses, as well as evaluating the possible specific aggressive behaviors, such as overt online aggressions or covert offline aggressive behaviors, to attest whether individual vulnerabilities can have a specific role in these maladaptive tendencies and the kind of the strengths of the associations [3,53]. 

Another limitation of our work is the limited sample size, which could influence the strength and the kind of the emerged associations [76]. Future research could benefit from larger sample sizes to test our hypothesis more precisely and accurately. Lastly, although we considered longitudinal predictions of impulsivity on offline and online aggressive behaviors, our time frame was short, as we considered only a 2-month interval to test our hypothesis. Therefore, our findings should be confirmed within a more comprehensive longitudinal framework to deeply analyze long-term associations among behavioral impulsivity, self-regulatory self-efficacy, and aggressive conduct in adolescence. In addition, future research could consider also other cultures, testing the cross-cultural validity and replicability of these results [81]. 

## 5. Conclusions

Despite several limitations of this study, our results took an important first step in that field of study that focuses on examining how temperamental vulnerabilities can predispose youths and adolescents to be more inclined to act in aggressive ways, offline and online [8,10,38,41]. 

In terms of theoretical implications, this work extended previous studies that focused on the effects of self-regulatory self-efficacy beliefs in offline contexts as a vehicle for contrasting aggressive and transgressive tendencies (see [53] for a review) and on the vulnerability of masculine populations to aggressive conduct [8]. Moreover, this work gave an important kickstart to re-write the conception of aggressive behaviors, offline and online, by evidencing how, nowadays, these problems do not only affect boys, so they should not still considered as “gendered” issues as in previous research, but a more comprehensive conceptualization of aggression should be examined, including the specific vulnerability aspects that could lead adolescent girls to be more inclined to aggressive responses, verbally and physically [26,27,34].

Our findings evidenced that impulsivity directly predicts more online aggression. However, adequate levels of self-regulatory self-efficacy can protect youths in online environments by affecting this relation because it indirectly predicts lower online aggression over time. In addition, our findings evidenced the protective role of self-efficacy in offline contexts because adequate levels of self-regulatory self-efficacy beliefs can mitigate the indirect effects of impulsivity impairments. In terms of practical and contextual implications, future intervention in aggressive behaviors should include a component related to enhancing youths’ self-regulatory self-efficacy, which was found to have a fundamental role in protecting young people from engaging in transgressive and maladaptive behaviors led by individual vulnerabilities [53]. According to a positive development approach [3,6,15,36], promoting general competencies and confidence, such as self-esteem, goal-oriented abilities, and self-efficacy in specific functioning domains, can improve preventive interventions’ effectiveness. In addition, future projects should consider how youths’ behaviors could be differently expressed according to the specific social context in which they are included, such as offline at-school interaction or online with-friends relations, to analyze more in-depth the similarities and differences of each context [39]. 

## Figures and Tables

**Figure 1 behavsci-14-00776-f001:**
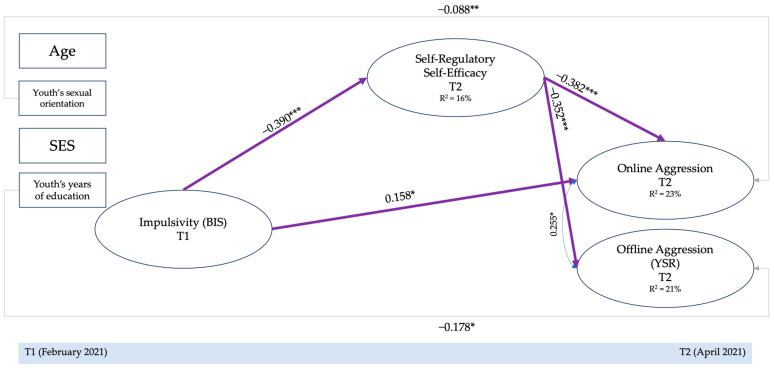
Mediation–moderation model: Full sample. Notes: Non-significant paths were estimated but not depicted. SES = socioeconomic status. Indirect effects: Impulsivity → Self-regulatory Self-efficacy → Online aggression = 0.149 ***. Impulsivity → Self-regulatory Self-efficacy → Offline aggression = 0.137 ***. * *p* < 0.050; ** *p* < 0.010; *** *p* < 0.001.

**Figure 2 behavsci-14-00776-f002:**
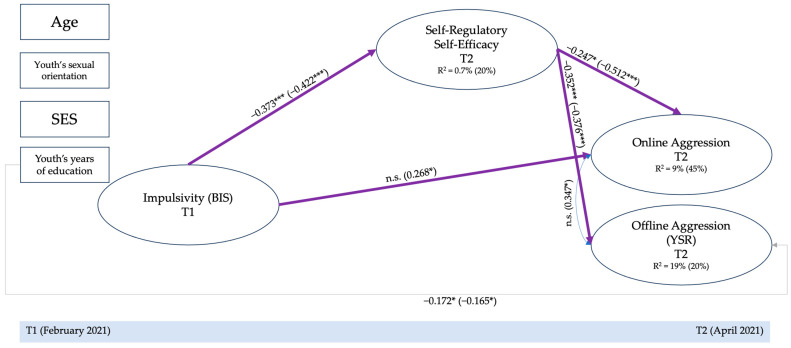
Mediation–moderation model: Multi-group by adolescents’ gender. Notes: Non-significant paths were estimated but not depicted. The first value refers to boys, while the second value in parenthesis refers to girls. Bold indicates different paths between boys and girls. Violet lines indicated equally estimated paths across the two genders, while green lines indicated freely estimated paths. SES = socio-economic status. Indirect effects: Impulsivity → Self-regulatory Self-efficacy → Online aggression = 0.092 ^n.s.^ (0.216 ***). Impulsivity → Self-regulatory Self-efficacy → Offline aggression = 0.131 ** (0.158 ***). * *p* < 0.050; ** *p* < 0.010; *** *p* < 0.001. n.s., not significant.

**Table 1 behavsci-14-00776-t001:** Multigroup by gender mediation–moderation model: Summary of goodness-of-fit statistics.

						Model Comparison				
	χ^2^	*Df*	Scal. Corr.	CFI	RMSEA		χ^2^ Diff	Δ *Df*	Δ CFI	Δ RMSEA
Full sample (*n* = 318)	358.21 *	295		0.97	0.03 [0.01–0.03]					
Model boys (*n* = 182)	400.37 ***	295		0.93	0.04 [0.03–0.05]					
Model girls (*n* = 128)	347.69 ^n.s.^	295		0.95	0.04 [0.02–0.05]					
Model 1. Free parameters	872.14 ***	628	1.07	0.91	0.05 [0.04–0.06]					
Model 2. Full constrained	912.12 ***	657	1.07	0.91	0.05 [0.04–0.06]	2 vs. 1	39.25 *	25	−0.01	0.00
Model 3. Partial constrained ^a^	902.63 ***	654	1.07	0.91	0.05 [0.04–0.06]	3 vs. 1	31.17	26	−0.01	0.00

Notes: *χ*^2^ = Chi-square goodness-of-fit; *Df* = degrees of freedom; Scal. Corr. = Scaling Correction Factor; CFI = Comparative Fit Index; RMSEA = Root-Mean-Square Error of Approximation. All Δ _index comparisons compared the model with the previous one were made. Partial constraints ^a^ = indirect effects of impulsivity on offline aggression; indirect effects of impulsivity on online aggression; correlation among the second and the fifth item of self-regulatory self-efficacy beliefs. * *p* < 0.050; *** *p* < 0.001. n.s., not significant.

**Table 2 behavsci-14-00776-t002:** Total, direct, and indirect effects of the two final models: summary of statistics.

Model	Dependent Variable	Total Effect		Direct Effect		Indirect Effect	
		Est.	*SE*	Est.	*SE*	Est.	*SE*
Full sample Model	Online Agg	0.308 ***	0.074	0.158 *	0.076	0.149 ***	0.044
	Offline Agg	0.226 **	0.069	0.089	0.067	0.137 ***	0.038
Boys Model	Online Agg	0.340 **	0.101	0.146	0.107	0.194 **	0.061
	Offline Agg	0.299 ***	0.088	0.184 *	0.091	0.115 *	0.048
Girls Model	Online Agg	0.437 ***	0.120	0.373 **	0.131	0.064	0.056
	Offline Agg	0.150	0.144	0.017	0.111	0.133 *	0.061
Multi-group Final Model	Online Agg	0.207 * (484 ***)	0.105 (0.085)	0.115 (0.268 *)	0.072 (0.107)	0.092 (0.216 ***)	0.051 (0.066)
	Offline Agg	0.205 ** (0.247 ***)	0.072 (0.073)	0.073 (0.088)	0.065 (0.074)	0.131 *** (0.158 ***)	0.041 (0.046)

Notes: Est. = Estimated path; SE = Standard Error; Online Agg = Online Aggression; Offline Agg = Offline Aggression. In the multi-group final model, the first value refers to boys, while the second value in parenthesis refers to girls. * *p* < 0.050; ** *p* < 0.010; *** *p* < 0.001.

## Data Availability

Research data and outputs of this study are openly available in the Open Science Framework (OSF) at https://doi.org/10.17605/OSF.IO/UACQ6 (accessed on 28 August 2024).

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
