# Peer review of "Online and Offline Aggressive Behaviors in Adolescence: The Role of Self-Regulatory Self-Efficacy Beliefs"

_behavsci, 2024, doi:10.3390/bs14090776_

Round 1

Reviewer 1 Report

Comments and Suggestions for Authors

In the current manuscript, the authors aim to test a moderated mediation model of the association between impulsivity, self-regulatory self-efficacy beliefs, and online/offline aggression. I commend the authors for studying online aggression, that is an important topic that deserves more attention in the literature. That being said, there are some issues that need to be addressed prior to publication:

1.      There are quite some language/grammar errors in the manuscript, e.g. “despite this topic was extensively studied” (line 49-50), “as the relative novelty of the issue” (line 52). Sometimes, words are also missing, e.g. […] “their emotional susceptibility increases, and activating self-regulating skills becomes more” à becomes more what? This makes the manuscript, and especially the introduction (which is quite long and wordy), difficult to read. Therefore, the authors might (1) consider consulting a native English speaker to proofread the manuscript and (2) try to make the introduction more clear by making it more brief and concise.

2.      Did you measure how much experience people have with regulating their behavior (online)? If self-efficay beliefs are “dynamic constructs that can be enhanced through mastery experiences as a result of individuals’ capacities to reflect and learn from experience”, that would be interesting to know.

3.      Did the school-based intervention influence any of the results you report here?

4.      Can you provide more information about the aggression measures?

a.      In what respects do they differ, apart from their offline and online nature? E.g. measuring overt/covert aggression, etc.

b.     Related to 4a, you do not seem to measure whether online aggression is indeed characterized differently than offline aggression (e.g. a-sync interactions, anonymity, etc), and the questionnaire also does not seem to take into account some of the new forms of aggression described in the introduction (e.g. cyber-stalking, shitstorms, trolling). I also wonder whether it would not have been more informative to measure aggressive behavior in the actual context (e.g. with experiments/as a state variable)? Could the authors address this limitation in the discussion / describe suggestions for future research?

c.      What is the association between the offline and online aggression measure?

d.     Do the authors think that it matters that they measures the two types of aggression on different time scales?

e.      Could you provide the mean scores for boys and girls separately for all scales of interest (online/offline aggression, self-efficacy beliefs)?

5.      The authors could offer a more substantial explanation for the absence of the association between impulsivity and offline aggression, especially since this is diverging so much from most empirical studies in the literature.

6.      The authors mention that their results are preliminary. Does that mean that they do not have enough power to statistically detect the effects they aim to test for in the study? It would be helpful if the authors clarify for which analyses they do have enough power (direct / indirect effects within their mediation model) and for which they probably do not (moderated mediation) (Xu et al., 2024); and perhaps this means they should refrain from interpreting/drawing conclusions from the moderated mediation analyses.

7.      Line 395 à it is unclear why you are talking about emotion regulation and delay discounting here; this is not what you measured.

8.      Sometimes Self-Regulatory Self-Efficacy Beliefs are considered a protective factor, and sometimes a risk-factor within the article – it might be more clear to stick to one of these perspectives.

References

Xu, Z., Gao, F., Fa, A., Qu, W., & Zhang, Z. (2024). Statistical power analysis and sample size planning for moderated mediation models. Behavior Research Methods. https://doi.org/10.3758/s13428-024-02342-2

Comments on the Quality of English Language

There are quite some language/grammar errors in the manuscript, e.g. “despite this topic was extensively studied” (line 49-50), “as the relative novelty of the issue” (line 52). Sometimes, words are also missing, e.g. […] “their emotional susceptibility increases, and activating self-regulating skills becomes more” à becomes more what? This makes the manuscript, and especially the introduction (which is quite long and wordy), difficult to read. Therefore, the authors might (1) consider consulting a native English speaker to proofread the manuscript and (2) try to make the introduction more clear by making it more brief and concise.

Author Response

1. There are quite some language/grammar errors in the manuscript, e.g. “despite this topic was extensively studied” (line 49-50), “as the relative novelty of the issue” (line 52). Sometimes, words are also missing, e.g. […] “their emotional susceptibility increases, and activating self-regulating skills becomes more” à becomes more what? This makes the manuscript, and especially the introduction (which is quite long and wordy), difficult to read. Therefore, the authors might (1) consider consulting a native English speaker to proofread the manuscript and (2) try to make the introduction more clear by making it more brief and concise.

OUR RESPONSE: We would first like to thank the reviewer for the accuracy of the reviewing work done. We revised the indicated lines suggested, and we carefully revised the entire manuscript to make it more fluid and comprehensible. We made the introduction briefer and clearer. We hope that our efforts improved the quality of the manuscript, following the reviewer's precious suggestion.

2. Did you measure how much experience people have with regulating their behavior (online)? If self-efficay beliefs are “dynamic constructs that can be enhanced through mastery experiences as a result of individuals’ capacities to reflect and learn from experience”, that would be interesting to know.

OUR RESPONSE: This is an interesting point for us. However, we did not measure “how much experience” participants had in regulating their behaviors, offline or online. This point could be an interesting future direction, so we included it in the discussion section (lines 459-466) and as a limitation of our study (lines 490-493).

3. Did the school-based intervention influence any of the results you report here?

OUR RESPONSE: No, the intervention did not influence any of our results. We specified it in paragraph 2.3 (page 6, lines 282-284).

4. Can you provide more information about the aggression measures?

a. In what respects do they differ, apart from their offline and online nature? E.g. measuring overt/covert aggression, etc.

b. Related to 4a, you do not seem to measure whether online aggression is indeed characterized differently than offline aggression (e.g. a-sync interactions, anonymity, etc), and the questionnaire also does not seem to take into account some of the new forms of aggression described in the introduction (e.g. cyber-stalking, shitstorms, trolling). I also wonder whether it would not have been more informative to measure aggressive behavior in the actual context (e.g. with experiments/as a state variable)? Could the authors address this limitation in the discussion / describe suggestions for future research?

c. What is the association between the offline and online aggression measure?

d. Do the authors think that it matters that they measures the two types of aggression on different time scales?

e. Could you provide the mean scores for boys and girls separately for all scales of interest (online/offline aggression, self-efficacy beliefs)?

OUR RESPONSE: Thank you for this comment, which allowed us to specify the kind of measures adopted deeply. Unfortunately, in our study, we did not measure aggressive behaviors in the actual context. We included a brief discussion on the limits of our measures in section 4.1 (lines 487-488, lines 493-501). We also briefly presented the major difference between the two measures in section 2.2.4. Regarding the associations between the two forms of aggression, in the total sample, we found a weak positive association (as reported in Figure 1; r = .255*), while in the multi-group by gender model, this association was significant only for girls (r = .347**; Figure 2). Regarding point d), we agreed with the suggestion provided by the reviewer, and we added a short explanation of it in the discussion section (lines 455-462). Lastly, in the appendix, we included a new Table (Table A2) to summarize the mean statistics of all the study variables separately for boys and girls.

5. The authors could offer a more substantial explanation for the absence of the association between impulsivity and offline aggression, especially since this is diverging so much from most empirical studies in the literature.

OUR RESPONSE: We revised the discussion on offline aggression, and we included a clearer explanation of the absence of a significant direct effect of impulsivity on offline aggressive conduct (lines 404-433).

6. The authors mention that their results are preliminary. Does that mean that they do not have enough power to statistically detect the effects they aim to test for in the study? It would be helpful if the authors clarify for which analyses they do have enough power (direct / indirect effects within their mediation model) and for which they probably do not (moderated mediation) (Xu et al., 2024); and perhaps this means they should refrain from interpreting/drawing conclusions from the moderated mediation analyses.

OUR RESPONSE: We considered our results as “preliminary,” not for statistical motivation but for theoretical and empirical motivation. In fact, to our knowledge, no previous studies specifically investigated concurrent associations between impulsivity and offline and online aggressive tendencies, analyzing the mediating role of self-regulatory self-efficacy beliefs, so our results should be confirmed and expanded by future research. If the reviewer believes that this specification could be relevant to the manuscript, we will include this in the conclusion section.

7. Line 395 à it is unclear why you are talking about emotion regulation and delay discounting here; this is not what you measured.

OUR RESPONSE: Thank you for this specification. We re-wrote the sentence to clarify why we included these two aspects of impulsivity (lines 415-416).

8. Sometimes Self-Regulatory Self-Efficacy Beliefs are considered a protective factor, and sometimes a risk-factor within the article – it might be more clear to stick to one of these perspectives.

OUR RESPONSE: Thank you for this comment. Throughout the entire manuscript, we made significant modifications and changes that hopefully clarify the role of self-efficacy in our study.

Reviewer 2 Report

Comments and Suggestions for Authors

The research analyzes the role of self-regulatory self-efficacy beliefs (i.e., SRSE) and their influence on the prevention of both offline and online aggressive behaviors. It is this technological and comparative aspect with respect to traditional aggression behaviors that gives the research a novel character that ensures the relevance of the study.

Both the title and the abstract are correct according to the study presented.

The introduction correctly establishes the research gap that justifies the study and conveniently defines the study variables, mentioning specific theoretical frameworks for both offline and online behaviors. In addition, it establishes the existing gender differences when considering the influence of impulsivity on online and offline aggressive behaviors and the role of self-regulatory self-efficacy beliefs. The objective and hypotheses are well defined but it would be convenient to include at the end of section 1.3. “The present Study” a numbered and concrete summary of the hypotheses. For example:

H1: High impulsivity would predict higher aggressive responses in both adolescent boys and girls.

H2: The protective role of self-regulatory self-efficacy beliefs would be stronger in boys than in girls.

H3, H4...

Method

In 2.1. the queer option is included as a sexual orientation, but actually queer status is not a sexual orientation, but rather an aspect related to sexual identity, so it is confusing.

The description of the sample, of the instruments for the statistical analyses is adequate, although the procedure for implementing the tests is not described and no mention is made of the ethical aspects of the research, which are usually included in the subsection on Procedure, within the method.

Results

The results are correctly explained in a sequential manner and no inconsistencies are apparent in the results.

Some of the results described in the text could be presented in tables for greater order.

Discussion

The interpretation of the results from the psychoeducational perspective and the comparison of results with other similar studies in the scientific literature are correct. However, the contrast of hypotheses by which these are confirmed or rejected is not exhaustively carried out. Taking into account what was pointed out in the introductory section on the specification of hypotheses, it would be appropriate in this section to mention whether each of them is confirmed or rejected.

Likewise, no mention has been made of the theoretical, practical and contextual implications of the study, a necessary aspect, and although some ways of overcoming the limitations of the study have been proposed, no progress has been made in the future lines of research that could anticipate future paths.

References

The bibliographic references are erroneous since the APA-7 italics are not included. In journal articles, the name of the journal and volume should be in italics. In monographs, the title, etc., should be in italics.

Author Response

1. The introduction correctly establishes the research gap that justifies the study and conveniently defines the study variables, mentioning specific theoretical frameworks for both offline and online behaviors. In addition, it establishes the existing gender differences when considering the influence of impulsivity on online and offline aggressive behaviors and the role of self-regulatory self-efficacy beliefs. The objective and hypotheses are well defined but it would be convenient to include at the end of section 1.3. “The present Study” a numbered and concrete summary of the hypotheses. For example:

H1: High impulsivity would predict higher aggressive responses in both adolescent boys and girls.

H2: The protective role of self-regulatory self-efficacy beliefs would be stronger in boys than in girls.

H3, H4...

OUR RESPONSE: Thank you for this suggestion, which, according to the other reviewer's comments, could make this paragraph clearer and concise. We modified section 1.3 by including a summary of our hypotheses (lines 175-193). Moreover, we included the reference to our specific hypotheses also in the discussion section to make the entire manuscript more fluid (lines 404-452).

2. Method: In 2.1. the queer option is included as a sexual orientation, but actually queer status is not a sexual orientation, but rather an aspect related to sexual identity, so it is confusing.

OUR RESPONSE: Thank you for this comment, which allowed us to detect this typo. We corrected the wrong “queer” word with the correct word (line 214).

3. Method: The description of the sample, of the instruments for the statistical analyses is adequate, although the procedure for implementing the tests is not described and no mention is made of the ethical aspects of the research, which are usually included in the subsection on Procedure, within the method.

OUR RESPONSE: In the first version of the manuscript, we dropped this information to make the section shorter, but we agreed that it was important to mention the crucial ethical aspects of data collection. These aspects were included in section 2.1 (lines 202-206).

4. Results: The results are correctly explained in a sequential manner and no inconsistencies are apparent in the results. Some of the results described in the text could be presented in tables for greater order.

OUR RESPONSE: Thank you for appreciating the way we explained our results. All the direct effects of the two models, together with the model fit information, were already provided in Table 1 and in Figures 1 and 2. The only missing information in tables or figures were the estimated indirect effects, which, to our knowledge, were usually provided directly in the text, as we did. However, following this suggestion, we included a second Table in the main text to summarize all the indirect effects included in the text (Table 2).

5. Discussion: The interpretation of the results from the psychoeducational perspective and the comparison of results with other similar studies in the scientific literature are correct. However, the contrast of hypotheses by which these are confirmed or rejected is not exhaustively carried out. Taking into account what was pointed out in the introductory section on the specification of hypotheses, it would be appropriate in this section to mention whether each of them is confirmed or rejected.

OUR RESPONSE: Thank you for allowing us to provide further specifications on our hypotheses. According to this comment, and as we specified in the previous response, we included these references to our hypotheses in the discussion to make the manuscript more fluid.

6. Likewise, no mention has been made of the theoretical, practical and contextual implications of the study, a necessary aspect, and although some ways of overcoming the limitations of the study have been proposed, no progress has been made in the future lines of research that could anticipate future paths.

OUR RESPONSE: Thank you for this suggestion. We included practical and theoretical implications in the conclusions section (lines 517-525, lines 532-542).

7. References: The bibliographic references are erroneous since the APA-7 italics are not included. In journal articles, the name of the journal and volume should be in italics. In monographs, the title, etc., should be in italics.

OUR RESPONSE: Thank you also for this accuracy. We revised all the references according to MDPI instructions, and we italicized all the names and volumes of the mentioned Journals.

Round 2

Reviewer 1 Report

Comments and Suggestions for Authors

The authors have handled the comments well and I now recommend this article for publication. 

Comments on the Quality of English Language

The authors clearly made some improvements in this regard.